

# Predicting central cervical lymph node metastasis in papillary thyroid carcinoma with Hashimoto's thyroiditis: a practical nomogram based on retrospective study

Lirong Wang[1,*], Lin Zhang[1,*], Dan Wang[1], Jiawen Chen[2], Wenxiu Su[3], Lei Sun[1], Jue Jiang[1], Juan Wang[1] and Qi Zhou[1]

[1] Department of Ultrasound, the Second Affiliated Hospital of Xi'an Jiaotong University, Xi'an, Shannxi, China
[2] Department of Otolaryngology-Head and Neck Surgery, the Second Affiliated Hospital of Xi'an Jiaotong University, Xi'an, Shannxi, China
[3] Department of Pathology, the Second Affiliated Hospital of Xi'an Jiaotong University, Xi'an, Shannxi, China
* These authors contributed equally to this work.

## ABSTRACT

**Background**. In papillary thyroid carcinoma (PTC) patients with Hashimoto's thyroiditis (HT), preoperative ultrasonography frequently reveals the presence of enlarged lymph nodes in the central neck region. These nodes pose a diagnostic challenge due to their potential resemblance to metastatic lymph nodes, thereby impacting the surgical decision-making process for clinicians in terms of determining the appropriate surgical extent.

**Methods**. Logistic regression analysis was conducted to identify independent risk factors associated with central lymph node metastasis (CLNM) in PTC patients with HT. Then a prediction model was developed and visualized using a nomogram. The stability of the model was assessed using ten-fold cross-validation. The performance of the model was further evaluated through the use of ROC curve, calibration curve, and decision curve analysis.

**Results**. A total of 376 HT PTC patients were included in this study, comprising 162 patients with CLNM and 214 patients without CLNM. The results of the multivariate logistic regression analysis revealed that age, Tg-Ab level, tumor size, punctate echogenic foci, and blood flow grade were identified as independent risk factors associated with the development of CLNM in HT PTC. The area under the curve (AUC) of this model was 0.76 (95% CI [0.71–0.80]). The sensitivity, specificity, accuracy, and positive predictive value of the model were determined to be 88%, 51%, 67%, and 57%, respectively.

**Conclusions**. The proposed clinic-ultrasound-based nomogram in this study demonstrated a favorable performance in predicting CLNM in HT PTCs. This predictive tool has the potential to assist clinicians in making well-informed decisions regarding the appropriate extent of surgical intervention for patients.

Corresponding author
Qi Zhou, 13909232905@163.com

## INTRODUCTION

In recent years, there has been a global increase in the incidence of papillary thyroid carcinoma (PTC) and Hashimoto's thyroiditis (HT), attracting significant clinical and epidemiological attention (*Latina et al., 2013*; *McLeod, Sawka & Cooper, 2013*). It has been reported that approximately one-third of PTC patients also have HT (*Antonelli et al., 2015*; *Caturegli, De Remigis & Rose, 2014*). Although PTC generally has a favorable prognosis, lymph node metastasis (LNM) continues to be a concern, affecting 40%–90% of PTC patients (*Wu et al., 2020*; *Zhong et al., 2022*). However, when PTC is accompanied by HT, the overall aggressiveness of PTC is reduced, leading to a lower incidence of LNM, decreased extrathyroidal extension, and a reduced risk of distant metastasis or recurrence (*Lai et al., 2015*). Specifically, HT PTCs exhibit a significantly lower rate of central lymph node metastasis (CLNM) compared to pure PTC, but no significant difference has been found in terms of lateral lymph node metastasis (*Xu et al., 2021*). Currently, there is limited research focusing on the detailed evaluation of CLNM in HT PTCs. Therefore, the development of a prediction model for CLNM in HT PTCs is urgently needed to assist clinicians in effectively managing these patients.

According to the 2015 American Thyroid Association guidelines (*Haugen et al., 2016*), patients with suspicious CLNM are recommended to undergo central compartment neck dissection during surgery. A large-scale meta-analysis found that prophylactic central compartment lymph node dissection in PTC patients reduces or shows no significant improvement in postoperative local recurrence rates; however, it is associated with decreased parathyroid function and a notable increase in recurrent laryngeal nerve injury rates (*Yang et al., 2023*). This underscores the importance of striking a balance in PTC patients—differentiating between benign and malignant lymph nodes and selectively clearing malignant nodes to reduce recurrence and minimize unnecessary surgical complications. However, distinguishing between inflammatory hyperplasia and metastatic lymph nodes in patients with PTC and HT is challenging (*Lai et al., 2015*; *Vita et al., 2018*). Ultrasound, the preferred imaging method for evaluating thyroid nodules and lymph nodes, has its limitations (*Ahn et al., 2008*). Ultrasonic identification of metastatic lymph nodes is more accurate when they exhibit local liquefaction or internal punctate echo foci (*Liu et al., 2021b*; *Xu et al., 2017*; *Zhao & Li, 2019*). However, distinguishing between metastatic and inflammatory hypertrophic lymph nodes is difficult due to shared features such as rounded shape and loss of fatty hilum (*Min et al., 2021*; *Xu et al., 2017*). Moreover, the complex anatomical structure of the central neck region results in a low sensitivity (approximately 33%) of ultrasound in identifying CLNM in PTC (*Zhao & Li, 2019*). This sensitivity is further reduced in HT PTCs due to the presence of lymph node inflammatory response hyperplasia. Therefore, there is a need for a prediction model independent of lymph node detection. Previous studies have shown that factors such as sex, age, tumor size, extrathyroidal extension, irregular margin, microcalcification, and taller-than-wider shape are closely associated with LNM in PTC (*Remonti et al., 2015*). However, the applicability of these factors in HT PTCs requires further evaluation.

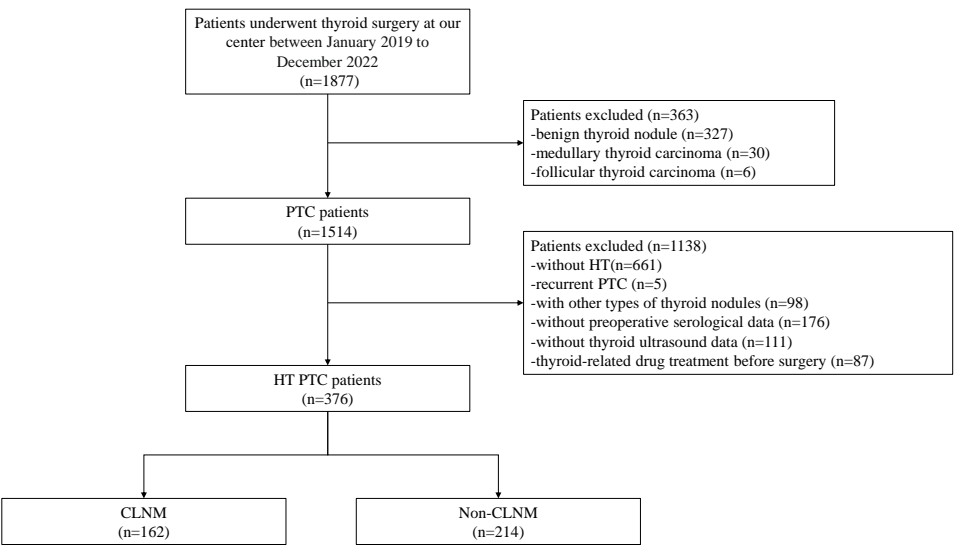

**Figure 1   Workflow of this study.**

This study aimed to analyze the clinical and ultrasonographic risk factors associated with CLNM in HT PTCs and develop a prediction model for CLNM that does not rely on lymph node detection. The study also provides a clinic-ultrasound-based nomogram, which can assist clinicians in their practice.

## MATERIALS & METHODS

### Patients

This retrospective study was approved by the Institutional Review Board of our center (2022259). Informed consent was waived due to the retrospective nature of the study. A total of 1,877 consecutive patients who underwent thyroidectomy at our center between January 2019 and December 2022 were included in the initial recruitment. Preoperative serological and thyroid ultrasonography were conducted for all patients, and the final postoperative histopathology confirmed the diagnoses. Exclusion criteria were applied, including patients without HT, patients with recurrent PTC, patients with other types of thyroid nodules, those with missing serological and ultrasound data, and patients who received preoperative radiation or oral medication. The inclusion and exclusion process are depicted in Fig. 1. After the application of these criteria, a total of 376 patients (43 males and 333 females) with an average age of 44.79 ± 11.72 were included in the study. Among them, 162 patients exhibited CLNM, while 214 patients did not.

### Clinical and pathological data

Prophylactic central lymph node dissection was performed at our center. In cases where lateral cervical lymph nodes were suspected to be positive, confirmation was obtained through ultrasound-guided fine needle puncture, followed by lymph node dissection in the corresponding area. Baseline data, such as age and sex, were collected prior to surgery.
The diagnosis of PTC and LNM was established based on postoperative histopathology. HT was diagnosed by evaluating postoperative histopathology as well as serum levels of thyroid peroxidase antibody (TPO-Ab) and thyroglobulin antibody (Tg-Ab). At our center, the normal reference ranges are as follows: thyroid stimulating hormone (TSH) is 0.27–4.2 μIU/ml, TPO-Ab is 0-34 IU/ml, and Tg-Ab is 0–115 IU/ml. Pathological findings, including maximum tumor diameter, multifocality, extrathyroidal extension, and LNM, were determined by the same team of pathologists.

## US Examination and image evaluation

Thyroid ultrasound examinations were conducted by radiologists with a minimum of 5 years of experience. Digital workstations were used to store images of thyroid lesions and cervical lymph nodes for subsequent evaluation. Two radiologists with at least 10 years of experience in thyroid disease diagnosis independently reviewed the ultrasound images. In case of any disagreement, a consensus was reached through consultation between the two radiologists.

The patients were positioned in the supine position to expose the neck fully and were instructed to breathe calmly. Transverse and longitudinal scans were meticulously performed to acquire conventional ultrasound images of the thyroid nodules. The ultrasound features of the nodules were recorded based on the TI-RADS classification, which included composition (mixed cystic and solid/solid or nearly solid), echogenicity (hyperechoic or isoechoic/hypoechoic or markedly hypoechoic), shape (wider-than-tall/taller-than-wide), margin (smooth or ill-defined/lobulated or irregular/with extrathyroidal extension), and presence of echogenic foci (none/macrocalcification or peripheral calcification/punctate echogenic foci). Furthermore, the blood flow signals measured by Doppler ultrasound were documented. The blood flow signals were categorized into four levels (0, 1, 2, or 3) according to Adler's semi-quantitative method (*Adler et al., 1990*). In cases of multifocal PTC, we documented the ultrasound features of the nodule with the highest TI-RADS classification. In instances where multiple nodules had the same classification, we recorded the ultrasound features of the largest diameter nodule.

Ultrasound examinations were conducted on the lateral and central neck regions of PTC patients to assess the overall lymph node metastasis situation. When lymph nodes exhibited ultrasound features such as irregular margins, microcalcifications, cystic changes, loss of lymph node hilum, or a long-axis/short-axis ratio <1.5 (*Yeh et al., 2015*), ultrasound suggested the potential presence of lymph node metastasis in the corresponding neck regions. If any lymph node displayed suspicious ultrasound features,this patient was classified as ultrasound-suspected LNM positive.

## Statistical analysis

For data that exhibited a skewed distribution, the median (first quartile, third quartile) was used for descriptive purposes. Other descriptive data were presented as mean ± standard deviation (SD). Statistical analysis was conducted using the independent samples $t$-test or Mann–Whitney U test. Enumeration data were expressed as percentages and

analyzed using the $\chi 2$ test or Fisher's exact test. Multivariate analysis was performed using logistic regression, and a nomogram was constructed based on the results. The DeLong test is employed to compare the performance of different ROC curves. The stability of the model was assessed using ten-fold cross-validation. The performance of the model was further evaluated using ROC curve analysis, calibration curve, and decision curve analysis. Statistical significance was defined as $p$-values less than 0.05. Data analysis was carried out using SPSS Statistics 20.0 software (Chicago, IL, USA), and the nomogram was generated using R software (R version 4.2.0; *R Core Team, 2022*).

## RESULTS

### Clinical and pathological characteristics of HT PTCs

A total of 376 HT PTC patients were included in this study, with 162 patients having CLNM and 214 patients not having CLNM. Patients with CLNM were found to be younger and had larger tumor sizes compared to those without CLNM, showing a significant difference ($p < 0.05$). However, no significant differences were observed in terms of sex, TSH levels, TPO-Ab levels, Tg-Ab levels, multifocality, and extrathyroidal extension ($p > 0.05$). It is noteworthy that among the HT PTC cases, ultrasound indicated suspicious CLNM in a total of 125 patients (93 cases Central only, 32 cases Central+ Lateral). However, postoperative pathology confirmed CLNM in only 64 patients (39 cases Central only, 25 cases Central+ Lateral), resulting in an accuracy rate of only 51.2% (Table 1).

### Ultrasonic characteristics of HT PTCs

Table 2 presents the ultrasonic characteristics of thyroid nodules in HT PTC patients. Among these characteristics, echogenic foci and the blood flow signals show significant differences between patients with CLNM and those without CLNM ($p < 0.05$). The CLNM group had a higher proportion of nodular punctate echogenic foci (76.5% *vs* 61.7%) and a lower proportion of macrocalcification or peripheral calcification (4.9% *vs* 7.9%), and absence of calcifications (18.5% *vs* 30.4%) compared to the non-CLNM group. Notably, nodules with CLNM predominantly exhibit a blood flow signal level of 0 (53.7%), and the incidence of CLNM decreases progressively with higher blood flow levels. Conversely, there are no significant differences observed in terms of nodule composition, echogenicity, shape, margin, and TI-RADS grading ($p > 0.05$).

### Univariate and multivariate analysis of CLNM in HT PTCs

According to the normal range of TSH levels at our center, patients were classified into three categories: normal thyroid function, hypothyroidism, and hyperthyroidism. Furthermore, based on the median values of TPO-Ab and Tg-Ab (82.30 IU/ml, 212.25 IU/ml), patients were categorized into high and low grades. Nodule margins were classified as either non-extrathyroidal extension or extrathyroidal extension, while echogenic foci were classified as non-punctate echogenic foci or punctate echogenic foci. In the univariate logistic regression analysis, variables such as age, Tg-Ab level, tumor size, punctate echogenic foci, and blood flow signals showed significant correlations with CLNM ($p < 0.05$). Additionally, sex and multifocality were included in the multivariate logistic regression analysis with marginal

**Table 1  Clinical and Pathological Characteristics of HT PTCs.**

| Characteristics | Total ($n = 376$) | CLNM ($n = 162$) | Non-CLNM ($n = 214$) | Statistic | p value |
|---|---|---|---|---|---|
| Sex, n(%) | | | | 3.21 | 0.073[a] |
| Male | 43(11.4) | 24(14.8) | 19(8.9) | | |
| Female | 333(88.6) | 138(85.2) | 195(91.1) | | |
| Age[*], years | 44.79 ± 11.72 | 41.10 ± 11.46 | 47.58 ± 11.16 | 5.51 | <0.001[b] |
| <55 | 299(79.5) | 145(98.5) | 154(72.0) | 17.43 | <0.001[a] |
| ≥55 | 77(20.5) | 17(10.5) | 60(28.0) | | |
| TSH[d], μIU/ml | 2.96 (1.88, 4.53) | 2.89 (1.87,4.43) | 3.04 (1.89,4.69) | 0.77 | 0.439[c] |
| TPO-Ab[d], IU/ml | 82.30 (19.06, 261.58) | 76.98 (19.23,243.78) | 88.18 (17.95,278.33) | 0.26 | 0.792[c] |
| Tg-Ab[d], IU/ml | 212.25 (64.35, 470.18) | 255.05 (63.70,535.25) | 172.05 (66.03,446.13) | 1.79 | 0.073[c] |
| Tumor size[d], mm | 8.90 (6.60, 12.80) | 10.95 (7.58,16.63) | 8.00 (6.20,10.13) | 6.17 | <0.001[c] |
| ≤10 | 231(61.4) | 71(43.8) | 160(74.8) | 37.25 | <0.001[a] |
| >10 | 145(38.6) | 91(56.2) | 54(25.2) | | |
| Multifocality,n (%) | | | | 2.92 | 0.093[a] |
| No | 259(68.9) | 104(64.2) | 155(72.4) | | |
| Yes | 117(31.1) | 58(35.8) | 59(27.6) | | |
| Extrathyroidal extension, n (%) | | | | 0.91 | 0.420[a] |
| No | 349(92.8) | 148(91.4) | 201(93.9) | | |
| Yes | 27(7.2) | 14(8.6) | 13(6.1) | | |
| Suspicious LNM_indicated by US, n (%) | | | | 16.19 | <0.001[a] |
| Positive | 151(40.2) | 84(51.9) | 67(31.3) | | |
| Negative | 225(59.8) | 78(48.1) | 147(68.7) | | |
| Positive type | | | | 18.40 | <0.001[a] |
| Only Central | 93(61.6) | 39(46.4) | 54(80.6) | | |
| Only Lateral | 26(17.2) | 20(23.8) | 6(9.0) | | |
| Central + Lateral | 32(21.2) | 25(29.8) | 7(10.4) | | |

**Notes.**
[a] $\chi^2$ test.
[b] Independent samples $T$-test.
[c] Mann–Whitney U test.
[*] The data is expressed as mean ± SD.
[d] The data is expressed as median (first quartile, third quartile).

significance ($0.05 < p < 0.1$). Ultimately, age, Tg-Ab level, tumor size, punctate echogenic foci, and blood flow grade were identified as independent risk factors for the development of CLNM in HT PTC ($p < 0.05$) (Table 3).

Figure 2 illustrates the receiver operating characteristic (ROC) curves of the CLNM prediction performance of different models for HT PTCs. The area under the curve (AUC) for the clinical-ultrasound model in predicting CLNM is 0.76 (95% CI [0.71–0.80]), while the AUC for the clinical feature model is 0.71 (95% CI [0.67–0.76]) , and the ultrasound feature model is 0.65 (95% CI [0.59–0.69]). The diagnostic performance of the clinical-ultrasound prediction model is significantly higher than that of the clinical model ($p = 0.047$) and the ultrasound model ($p = 0.002$). The AUC of the clinical model is higher than that of the ultrasound model, but the difference is not statistically significant ($p = 0.069$). Additionally, the clinical-ultrasound prediction model's Youden index was
**Table 2 Ultrasonic characteristics of HT PTCs.**

| Characteristics, n (%) | Total ($n = 376$) | CLNM ($n = 162$) | Non-CLNM ($n = 214$) | Statistic | $p$ value |
|---|---|---|---|---|---|
| Composition | | | | / | 0.656[b] |
| Mixed cystic and solid | 5(1.3) | 3(1.9) | 2(0.9) | | |
| Solid or almost solid | 371(98.7) | 159(98.1) | 212(99.1) | | |
| Echogenicity | | | | 0.05 | 1.000[a] |
| Hyperechoic or isoechoic | 13(3.5) | 6(3.7) | 7(3.3) | | |
| Hypoechoic or very hypoechoic | 363(96.5) | 156(96.3) | 207(96.7) | | |
| Shape | | | | 1.96 | 0.173[a] |
| Wider-than-tall | 212(56.4) | 98(60.5) | 114(53.3) | | |
| Taller-than-wide | 164(43.6) | 64(39.5) | 100(46.7) | | |
| Margin | | | | 2.18 | 0.337[a] |
| Smooth or ill-defined | 149(39.6) | 61(37.7) | 88(41.1) | | |
| Lobulated or irregular | 186(49.5) | 79(48.8) | 107(50.0) | | |
| Extra-thyroidal extension | 41(10.9) | 22(13.5) | 19(8.9) | | |
| Echogenic Foci | | | | 9.37 | 0.009[a] |
| None | 95(25.3) | 30(18.5) | 65(30.4) | | |
| Macrocalcification or peripheral calcification | 25(6.6) | 8(4.9) | 17(7.9) | | |
| Punctate echogenic foci | 256(68.1) | 124(76.5) | 132(61.7) | | |
| Blood Flow | | | | 16.77 | <0.001[a] |
| 0 | 161(42.8) | 87(53.7) | 74(34.6) | | |
| 1 | 150(39.9) | 56(34.6) | 94(43.9) | | |
| 2 | 55(14.6) | 18(11.1) | 37(17.3) | | |
| 3 | 10(2.7) | 1(0.6) | 9(4.2) | | |
| TI-RADS | | | | 1.61 | 0.498[b] |
| 3 | 4(1.1) | 3(1.9) | 1(0.5) | | |
| 4 | 73(19.4) | 31(19.1) | 42(19.6) | | |
| 5 | 299(79.5) | 128(79.0) | 171(79.9) | | |

**Notes.**
[a] $\chi^2$ test.
[b] Fisher's exact test.

found to be 0.38; the corresponding cut-off value was determined as $-0.78$, resulting in a sensitivity (SEN) of 88%, specificity (SPE) of 51%, positive predictive value (PPV) of 57%, negative predictive value (NPV) of 84%, and overall accuracy (ACC) of 67% in predicting CLNM in HT PTCs.

The equation of the clinical-ultrasound prediction model is as followed:

Logit $(\pi) = -1.40-1.27 \times$ age $+ 0.49 \times$ TG-Ab $+ 1.30 \times$ tumor size $+ 0.74 \times$ calcification $-0.64 \times$ blood(1) $-0.93 \times$ blood(2) $-2.14 \times$ blood(3). $\pi$ means the conditional probability of CLNM.

## Construction of clinic-ultrasound-based nomogram and validation

A nomogram (Fig. 3) was constructed based on the results of the multivariate logistic regression analysis, incorporating five variables: age, Tg-Ab level, tumor size, punctate echogenic foci, and blood flow grade. This nomogram serves as a visual tool for clinicians to predict the likelihood of CLNM in HT PTCs. An example of the prediction model is

**Table 3  Predictive Factors for CLNM of HT PTCs.**

| Variables of CLNM | Univariate analysis | | | Multivariate analysis | | |
|---|---|---|---|---|---|---|
| | Crude OR | 95% CI | *p* value | Adjusted OR | 95% CI | *p* value |
| Sex (Female/Male) | 1.79 | 0.94–3.39 | 0.076 | | | 0.124 |
| Age, years (<55/≥55) | 0.30 | 0.17–0.54 | <0.001 | 0.28 | 0.15–0.53 | *<0.001* |
| Thyroid function | | | 0.523 | | | |
| (Normal/ | Reference | | | | | |
| Hypothyroidism/ | 0.82 | 0.51–1.30 | 0.386 | | | |
| Hyperthyroidism) | 0.70 | 0.32–1.54 | 0.378 | | | |
| TPO-Ab (Low/High) | 0.92 | 0.61–1.38 | 0.677 | | | |
| Tg-Ab (Low/High) | 1.69 | 1.12–2.55 | 0.013 | 1.64 | 1.03–2.59 | *0.035* |
| Tumor size, mm (≤10/>10) | 3.80 | 2.45–5.88 | <0.001 | 3.68 | 2.30–5.87 | *<0.001* |
| Multifocality (No/Yes) | 1.47 | 0.94–2.27 | 0.088 | | | 0.583 |
| Extrathyroidal extension (No/Yes) | 1.46 | 0.67–3.20 | 0.342 | | | |
| Composition (Mixed cystic and solid/Solid or almost solid) | 0.50 | 0.08–3.03 | 0.451 | | | |
| Echogenicity (Hyperechoic or isoechoic/Hypoechoic or very hypoechoic) | 0.88 | 0.29–2.67 | 0.879 | | | |
| Shape (Wider-than-tall/Taller-than-wide) | 0.74 | 0.49–1.13 | 0.162 | | | |
| Margin (Non-extrathyroidal extension/Extrathyroidal extension) | 1.61 | 0.84–3.09 | 0.150 | | | |
| Echogenic Foci (Non-punctate echogenic foci/Punctate echogenic foci) | 2.03 | 1.29–3.20 | 0.002 | 2.11 | 1.27–3.49 | *0.004* |
| Blood Flow (0/ | Reference | | 0.002 | | | *0.007* |
| 1/ | 0.51 | 0.32–0.80 | 0.003 | 0.53 | 0.32–0.87 | *0.012* |
| 2/ | 0.41 | 0.22–0.79 | 0.007 | 0.39 | 0.19–0.80 | *0.010* |
| 3) | 0.10 | 0.01–0.76 | 0.027 | 0.12 | 0.01–1.10 | 0.060 |
| TI-RADS (3/ | Reference | | | | | |
| 4/ | 0.25 | 0.02–2.48 | 0.234 | | | |
| 5) | 0.25 | 0.03–2.43 | 0.250 | | | |

illustrated in Fig. 4. The performance of the model was validated using ten-fold cross-validation, resulting in an average AUC of 0.74 (Fig. 5).

To assess the reliability and practicality of the nomogram, the calibration curve (Fig. 6A) demonstrates a favorable consistency between the predicted probability of CLNM and the confirmed pathological probability, with an average absolute error of 0.018. The clinical decision curve analysis (DCA) curve (Fig. 6B) shows that the nomogram provides the optimal net benefit for predicting CLNM of HT PTCs when the threshold probability falls within the range of 10% to 76%. These findings highlight the effectiveness and utility of the nomogram as a predictive tool for CLNM in HT PTCs.

## DISCUSSION

In this study, we successfully developed and validated a nomogram based on clinical and ultrasound features to predict CLNM in preoperative HT PTCs. The nomogram exhibited excellent discriminative ability, with a maximum AUC of 0.83, surpassing the subjective

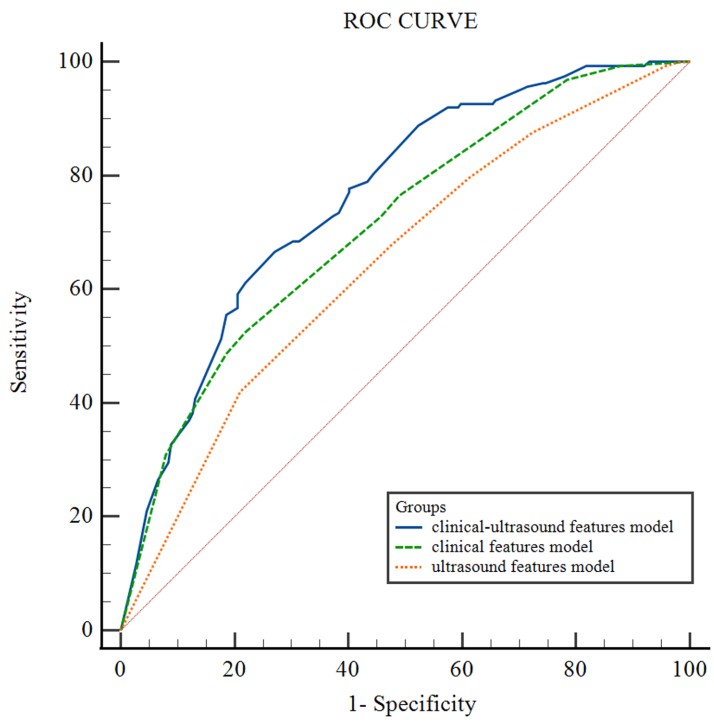

**Figure 2** ROC curve of different HT PTCs' CLNM prediction models.

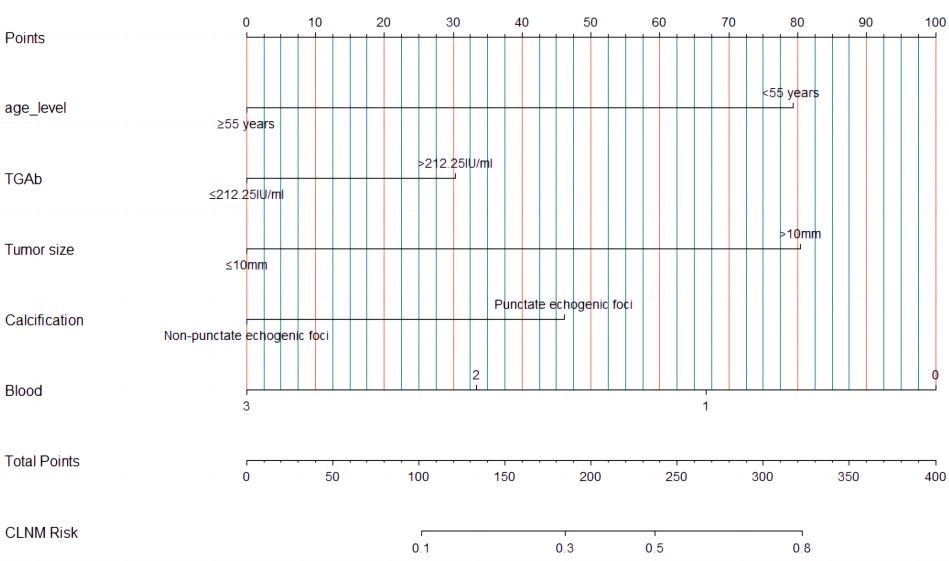

**Figure 3** A multiparametric nomogram predicting the CLNM of HT PTCs.

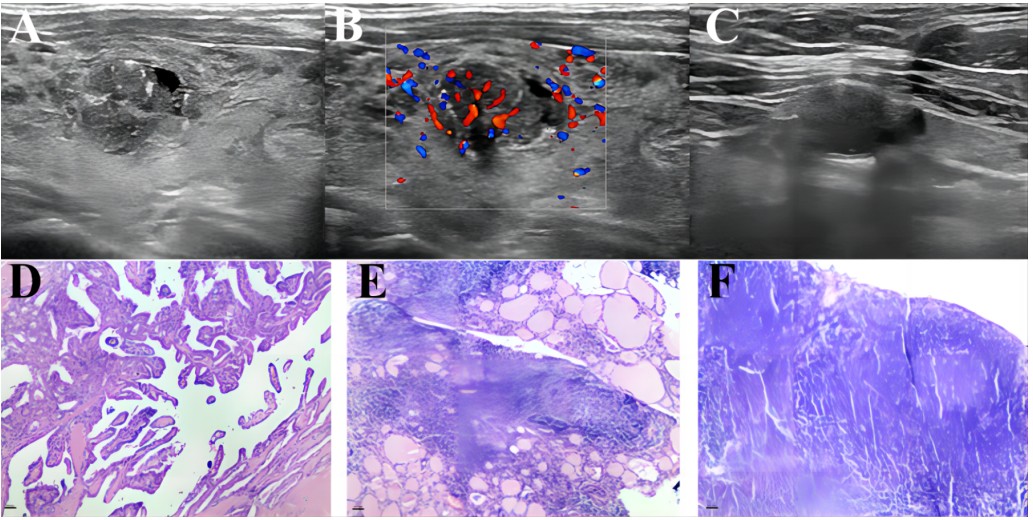

**Figure 4** **An example of a presentation using our nomogram to predict CLNM of HT PTC.** A 45-year-old male patient with a 13 ×11 mm hypoechoic nodule in the left lobe of the thyroid was pathologically confirmed to have PTC with HT and no CLNM. Preoperative serological examination of the patient indicated TSH, 6.85 μIU/ml; TPO-Ab, 291.1 UI/ml; Tg-Ab, 59.56 UI/ml. (A–B) The maximum diameter of the tumor was 13 mm, with punctate echogenic foci, and the blood flow grade was 3. (C) Lymph node enlargement in the central region of the neck, with ultrasound suggesting suspected metastasis. (D–F) showed pathological images of PTC, HT and proliferative lymph nodes of this patients respectively. According to the nomogram above, the total score of this patient was 206 (79 + 0 + 81 + 46 + 0), and the probability of CLNM was about 0.39. The magnification of HE staining was 40 × (eyepiece 4 ×, light lens 10 ×).

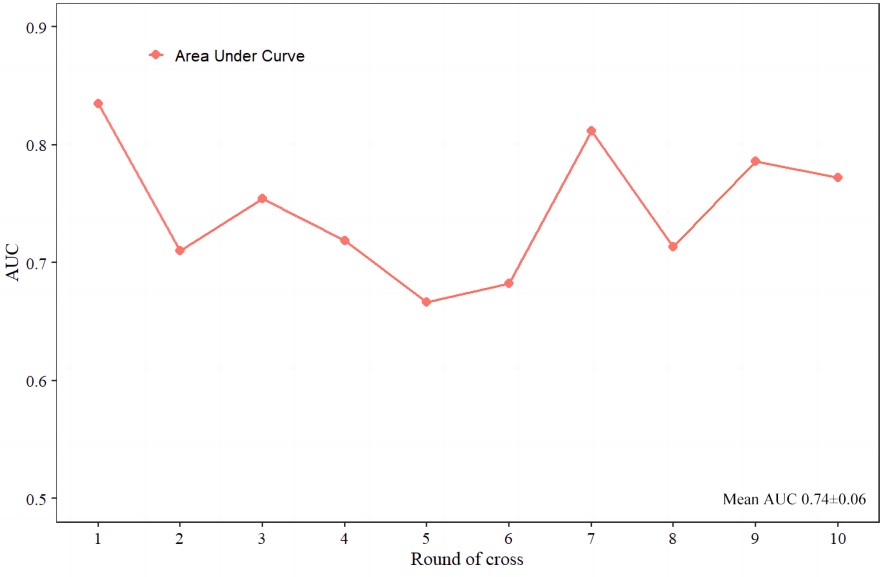

**Figure 5** **The ten-fold cross-validation of the predictive model.**

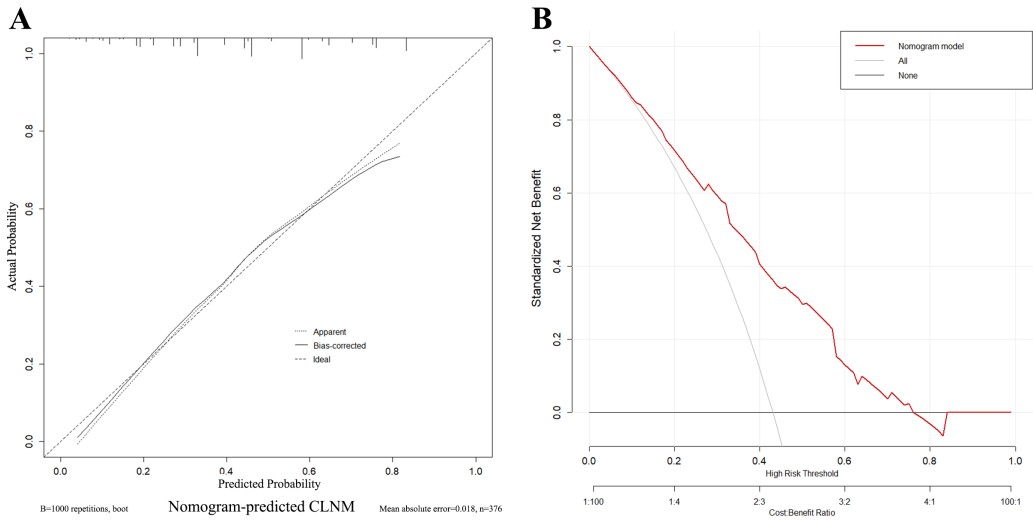

**Figure 6   Calibration curves and decision curve of the nomogram.** (A) Different types of lines display consistency between predicted and actual probabilities of CLNM. (B) The *X*-axis represents the threshold probability, and the *Y*-axis represents the net benefit. The decision curve demonstrates that when the threshold probability is within the range of 10%–76%, the nomogram for predicting the CLNM of HT PTCs brings more benefits than treating all or none of the patients.

predictions made by radiologists. Calibration curves demonstrated strong agreement between predicted and actual probabilities. Furthermore, the nomogram demonstrated the highest net benefit within a specific threshold probability range. This user-friendly predictive model has the potential to improve clinical decision-making for HT PTCs, providing valuable support and optimizing patient management.

High-resolution ultrasound is the preferred imaging modality for evaluating cervical lymph node metastasis in PTC (*Rago & Vitti, 2022*). However, the sensitivity of ultrasound in detecting PTC CLNM is relatively low, ranging from 28% to 33%, mainly due to the complex anatomical structures in the central neck region (*Xing et al., 2020*; *Zhao & Li, 2019*). Moreover, in the presence of HT, the accuracy of ultrasound in detecting CLNM is further compromised by the presence of inflammatory lymphadenopathy associated with HT (*Liu et al., 2021a*). In this study, we observed that radiologists identified suspicious CLNM in 40.2% of HT PTC cases, but only 55.6% of these cases were confirmed to have CLNM upon postoperative pathology. Conversely, among patients with no CLNM indicated by ultrasound, 34.7% still exhibited CLNM. These results are consistent with previous findings and suggest that PTC patients with HT are more prone to false-positive indications of CLNM (*Li et al., 2022*). This not only affects the accuracy of ultrasound diagnosis but also has implications for surgical decision-making. However, most previous studies have only considered HT as a general factor predicting CLNM in PTC, without conducting a detailed analysis of the specific clinical and ultrasound factors that impact CLNM in HT PTCs (*Lee et al., 2013*; *Moon et al., 2018*; *Tang, Pan & Peng, 2022*; *Xu et al., 2021*). Therefore, there is a need for comprehensive research focusing on the specific

factors that influence CLNM in HT PTCs, in order to improve diagnostic accuracy and assist clinicians in making informed decisions regarding the extent of surgical intervention.

The results of the univariate and multivariate logistic regression analyses in this study identified several independent predictive factors for CLNM in HT PTCs. These factors include age, TG-Ab levels, tumor size, calcifications, and blood flow grade. Indeed, most studies have identified age, sex, tumor size, multifocality, calcifications, and extrathyroidal extension as independent predictive factors for PTC CLNM (*Guang et al., 2021*; *Li et al., 2022*; *Li et al., 2021*; *Liu et al., 2019*; *Zhang et al., 2022*). These findings indicate that there is heterogeneity in the clinical and ultrasound features between HT PTCs and PTCs without HT. Age, tumor size, and calcifications have consistently been identified as independent predictive factors for CLNM in both HT PTCs and PTCs without HT, which is in line with previous studies. However, high TG-Ab levels (>212.25 IU/ml) were specifically associated with predicting CLNM in HT PTCs. Similar findings have been reported by other studies as well (*Aydoğan et al., 2021*; *Xu et al., 2023*).

Interestingly, we observed that as the blood flow grade of tumor nodules increased, the rate of CLNM tended to decrease in HT PTCs. This finding contrasts with the results of a study by Yang et al. (*Guang et al., 2021*), who found that abundant intraocular blood flow was a risk factor for LNM in PTCs. We speculate that in the presence of HT, the infiltrating lymphocytes associated with the tumor may differentiate into an anti-tumor phenotype under higher blood flow grades, thereby counteracting tumor growth. On the other hand, in the absence of HT, abundant intraocular blood flow may indicate rapid tumor growth, leading to an increased risk of LNM. In this study, the influence of the patient's sex on CLNM in HT PTCs was no longer significant. This may be due to the predominance of HT PTCs in females or a potential protective effect of HT in inhibiting CLNM, particularly in males. Additionally, certain aggressive features such as multifocality, extrathyroidal extension, and taller-than-wide nodules were no longer identified as independent risk factors in HT PTCs. These findings align with previous research (*Lee et al., 2013*; *Moon et al., 2018*; *Tang, Pan & Peng, 2022*; *Xu et al., 2021*) and may suggest a protective effect of HT, reducing the invasiveness of PTC. Overall, the results of this study provide insights into the unique factors associated with CLNM in HT PTCs and highlight the importance of considering HT status when predicting CLNM in PTC patients.

Based on the multivariate logistic regression analysis, a predictive model for preoperative HT PTCs' CLNM was constructed and visualized. To ensure the stability and reliability of the model, ten-fold cross-validation was performed using all patients. The model demonstrated good performance with an AUC of 0.76 on the ROC curve. It also showed a high sensitivity (88%) and negative predictive value (84%), indicating its ability to correctly identify patients without CLNM. However, it is worth noting that the model had low specificity (51%), positive predictive value (57%), and overall accuracy (67%). This suggests that the model may have a higher rate of false positives, leading to a lower specificity and positive predictive value. Therefore, in the clinical application of this model, clinicians should exercise caution and control the threshold probability within the range of 10% to 76% to maximize the benefit for patients.

Our study still has certain limitations. Firstly, as a single-center retrospective study, selection bias cannot be completely avoided. We mitigated this bias by continuously collecting data on HT PTC patients. Secondly, our study lacks external validation, and there is an urgent need for multicenter data to validate the generalizability and reliability of the model. Thirdly, the ultrasound data relied on interpretations from radiologists based on two-dimensional images, which may introduce subjectivity and limitations. In the future research, we will utilize Artificial Intelligence to extract multi-modal imaging omics features from tumors and establish deep learning models to optimize the performance of the predictive model.

## CONCLUSION

Age < 55 years, TG-Ab > 212.25 IU/ml, tumor size ≥10 mm, intratumoral punctate echogenic foci, and low blood flow grade are independent risk factors for HT PTCs CLNM. Our nomogram based on these factors can assist clinicians in optimizing treatment decisions for HT PTCs.

### Funding

This work was supported by the National Natural Science Foundation of China (No.82202183) and the Key R&D Program of Shaanxi Province (2023-YBSF-392). The funders had no role in study design, data collection and analysis, decision to publish, or preparation of the manuscript.

### Grant Disclosures

The following grant information was disclosed by the authors:
National Natural Science Foundation of China: 82202183.
Key R&D Program of Shaanxi Province: 2023-YBSF-392.

### Competing Interests

The authors declare there are no competing interests.

### Author Contributions

- Lirong Wang conceived and designed the experiments, performed the experiments, analyzed the data, prepared figures and/or tables, and approved the final draft.
- Lin Zhang conceived and designed the experiments, performed the experiments, analyzed the data, prepared figures and/or tables, and approved the final draft.
- Dan Wang performed the experiments, analyzed the data, prepared figures and/or tables, and approved the final draft.
- Jiawen Chen performed the experiments, analyzed the data, prepared figures and/or tables, and approved the final draft.
- Wenxiu Su analyzed the data, prepared figures and/or tables, and approved the final draft.

- Lei Sun conceived and designed the experiments, authored or reviewed drafts of the article, and approved the final draft.
- Jue Jiang conceived and designed the experiments, authored or reviewed drafts of the article, and approved the final draft.
- Juan Wang conceived and designed the experiments, authored or reviewed drafts of the article, and approved the final draft.
- Qi Zhou conceived and designed the experiments, authored or reviewed drafts of the article, and approved the final draft.

### Human Ethics

The following information was supplied relating to ethical approvals (*i.e.*, approving body and any reference numbers):

The Institutional Review Board of the Second Affiliated Hospital of Xi'an Jiaotong University

### Data Availability

The raw data is available in the Supplemental Files.

### Supplemental Information

Supplemental information for this article can be found online at http://dx.doi.org/10.7717/peerj.17108#supplemental-information.

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
