# Peer review of "Predicting central cervical lymph node metastasis in papillary thyroid carcinoma with Hashimoto’s thyroiditis: a practical nomogram based on retrospective study"

_PeerJ, doi:10.7717/peerj.17108_

## Round 0.1 · original submission · Minor Revisions

Reviewers have provided feedback on both the experimental design and findings in the manuscript. We request that you revise the manuscript in accordance with the received feedback, as this will contribute to elevating the standards of the manuscript. Thanks

·

Basic reporting

First of all, thank you very much to the author for giving me the opportunity to read about such an interesting topic. But some contents need to be modified to make the manuscript more perfect.

Experimental design

1.The distinction between metastatic and benign lymph nodes is indeed a challenge.But for patients with papillary thyroid cancer, does it make sense to distinguish whether the central lymph node metastases or not? As far as I know, central lymph node dissection is a necessary procedure for patients with papillary thyroid cancer. Please cite the literature to support your opinion.
2.Please elaborate on how to confirm that the lymph nodes observed by ultrasound are the positive lymph nodes dissected during the operation. How to achieve one-to-one correspondence? You know, there's often more than one central lymph node.
3.The study included 376 patients, right? Each patient had a single thyroid lesion? And in line 238 it says "multifocal is not identified as an independent risk factor for HT PTC." Please specify in the article.
4.Please mark the multiples of light mirror and eyepiece in the pathological pictures shown in Figure 4.
5.Although the model has certain value in identifying central lymph nodes in patients with HT PTC, the AUC value is only 0.76, the specificity is only 51%, and the positive predictive value is only 57%, which is not very perfect. It is recommended to compare the model with a number of doctors with different experience to better illustrate the advantages of the model.

Validity of the findings

6.Although the correction curve is close to perfect, the diagnostic efficiency of the model is not perfect, and it is recommended to further verify it in an external validation set to eliminate overfitting.
7.Is there any theoretical basis or literature reference for dividing the age limit by 55 years in manuscript?

Reviewer 2 ·

Basic reporting

The article uses clear, unambiguous and technically correct English expression that meets professional standards of courtesy and express.And the structure of the article conform to an acceptable format of ‘standard sections.However, the expression format of central lymph node metastasis in this abstract is incorrect, and the first occurrence of "CLNM" is not written according to the standard format of central lymph node metastasis (CLNM).

Experimental design

The clinical problems solved in this study are clear, and the clinical ultrasound based nomogram proposed in this study has a good performance in predicting the CLNM of HT ptc. This predictive tool has the potential to help clinicians make informed decisions about the appropriate level of surgical intervention for patients, is relevant and meaningful, The experimental design is reasonable and logical,but is no more innovative than previous studies.
The author mentioned in the article that previous studies have shown that factors such as sex, age, tumor size, extrathyroidal extension, irregular margin, microcalcification, and taller-than-wider shape are closely associated with LNM in PTC, but the male-female ratio in the patients included in this study was almost 1:8, indicating a serious imbalance in the gender ratio.

Validity of the findings

In the conclusion, the author pointed out that age, TG-Ab, tumor size, intratumoral stipple echo foci, and blood flow grade are independent risk factors for the occurrence of CLNM in HT ptc. Previous studies on these issues by others may have pointed out ,whether more innovative and advanced conclusions can be explored in this study.

·

Basic reporting

No Comment

Experimental design

No comment

Validity of the findings

10-fold cross validation had been done.

Additional comments

No comments

---

## Round 0.2 · accepted · Accept

The authors incorporated revisions based on the feedback from reviewers, modifying both the experimental design and findings in the manuscript. This impact is particularly meaningful in the context of thyroid cancer research.

·

Basic reporting

No Comment

Experimental design

No Comment

Validity of the findings

No Comment